# RESIDUAL CONNECTIONS ENCOURAGE ITERATIVE INFERENCE

**Stanisław Jastrzębski**[1,2,*]**, Devansh Arpit**[2,*]**, Nicolas Ballas**[3]**, Vikas Verma**[5]**,
Tong Che**[2] **& Yoshua Bengio**[2,6]

[1] Jagiellonian University, Cracow, Poland
[2] MILA, Université de Montréal, Canada
[3] Facebook, Montreal, Canada
[4] University of Bonn, Bonn, Germany
[5] Aalto University, Finland
[6] CIFAR Senior Fellow
[*] Equal Contribution

## ABSTRACT

Residual networks (Resnets) have become a prominent architecture in deep learning. However, a comprehensive understanding of Resnets is still a topic of ongoing research. A recent view argues that Resnets perform iterative refinement of features. We attempt to further expose properties of this aspect. To this end, we study Resnets both analytically and empirically. We formalize the notion of iterative refinement in Resnets by showing that residual connections naturally encourage features of residual blocks to move along the negative gradient of loss as we go from one block to the next. In addition, our empirical analysis suggests that Resnets are able to perform both representation learning and iterative refinement. In general, a Resnet block tends to concentrate representation learning behavior in the first few layers while higher layers perform iterative refinement of features. Finally we observe that sharing residual layers naively leads to representation explosion and counterintuitively, overfitting, and we show that simple existing strategies can help alleviating this problem.

## 1 INTRODUCTION

Traditionally, deep neural network architectures (e.g. VGG Simonyan & Zisserman (2014), AlexNet Krizhevsky et al. (2012), etc.) have been compositional in nature, meaning a hidden layer applies an affine transformation followed by non-linearity, with a different transformation at each layer. However, a major problem with deep architectures has been that of vanishing and exploding gradients. To address this problem, solutions like better activations (ReLU Nair & Hinton (2010)), weight initialization methods Glorot & Bengio (2010); He et al. (2015) and normalization methods Ioffe & Szegedy (2015); Arpit et al. (2016) have been proposed. Nonetheless, training compositional networks deeper than $15 - 20$ layers remains a challenging task.

Recently, residual networks (Resnets He et al. (2016a)) were introduced to tackle these issues and are considered a breakthrough in deep learning because of their ability to learn very deep networks and achieve state-of-the-art performance. Besides this, performance of Resnets are generally found to remain largely unaffected by removing individual residual blocks or shuffling adjacent blocks Veit et al. (2016). These attributes of Resnets stem from the fact that residual blocks transform representations additively instead of compositionally (like traditional deep networks). This additive framework along with the aforementioned attributes has given rise to two school of thoughts about Resnets– the ensemble view where they are thought to learn an exponential ensemble of shallower models Veit et al. (2016), and the unrolled iterative estimation view Liao & Poggio (2016); Greff et al. (2016), where Resnet layers are thought to iteratively refine representations instead of learning new ones. While the success of Resnets may be attributed partly to both these views, our work takes

steps towards achieving a deeper understanding of Resnets in terms of its iterative feature refinement perspective. Our contributions are as follows:

1. We study Resnets analytically and provide a formal view of iterative feature refinement using Taylor's expansion, showing that for any loss function, a residual block naturally encourages representations to move along the negative gradient of the loss with respect to hidden representations. Each residual block is therefore encouraged to take a gradient step in order to minimize the loss in the hidden representation space. We empirically confirm this by measuring the cosine between the output of a residual block and the gradient of loss with respect to the hidden representations prior to the application of the residual block.

2. We empirically observe that Resnet blocks can perform both hierarchical representation learning (where each block discovers a different representation) and iterative feature refinement (where each block improves slightly but keeps the semantics of the representation of the previous layer). Specifically in Resnets, lower residual blocks learn to perform representation learning, meaning that they change representations significantly and removing these blocks can sometimes drastically hurt prediction performance. The higher blocks on the other hand essentially learn to perform iterative inference– minimizing the loss function by moving the hidden representation along the negative gradient direction. In the presence of shortcut connections[1], representation learning is dominantly performed by the shortcut connection layer and most of residual blocks tend to perform iterative feature refinement.

3. The iterative refinement view suggests that deep networks can potentially leverage intensive parameter sharing for the layer performing iterative inference. But sharing large number of residual blocks without loss of performance has not been successfully achieved yet. Towards this end we study two ways of reusing residual blocks: 1. Sharing residual blocks during training; 2. Unrolling a residual block for more steps that it was trained to unroll. We find that training Resnet with naively shared blocks leads to bad performance. We expose reasons for this failure and investigate a preliminary fix for this problem.

## 2 BACKGROUND AND RELATED WORK

**Residual Networks and their analysis**:

Recently, several papers have investigated the behavior of Resnets (He et al., 2016a). In (Veit et al., 2016; Littwin & Wolf, 2016), authors argue that Resnets are an ensemble of relatively shallow networks. This is based on the unraveled view of Resnets where there exist an exponential number of paths between the input and prediction layer. Further, observations that shuffling and dropping of residual blocks do not affect performance significantly also support this claim. Other works discuss the possibility that residual networks are approximating recurrent networks (Liao & Poggio, 2016; Greff et al., 2016). This view is in part supported by the observation that the mathematical formulation of Resnets bares similarity to LSTM (Hochreiter & Schmidhuber, 1997), and that successive layers cooperate and preserve the feature identity. Resnets have also been studied from the perspective of boosting theory Huang et al. (2017). In this work the authors propose to learn Resnets in a layerwise manner using a local classifier.

Our work has critical differences compared with the aforementioned studies. Most importantly we focus on a precise definition of iterative inference. In particular, we show that a residual block approximate a gradient descent step in the activation space. Our work can also be seen as relating the gap between the boosting and iterative inference interpretations since having a residual block whose output is aligned with negative gradient of loss is similar to how gradient boosting models work.

**Iterative refinement and weight sharing**:

Humans frequently perform predictions with iterative refinement based on the level of difficulty of the task at hand. A leading hypothesis regarding the nature of information processing that happens in the visual cortex is that it performs fast feedforward inference (Thorpe et al., 1996) for easy stimuli or when quick response time is needed, and performs iterative refinement of prediction for complex

---

[1]A shortcut connection is a convolution layer between residual blocks useful for changing the hidden space dimension (see He et al. (2016a) for instance).

stimuli (Vanmarcke et al., 2016). The latter is thought to be done by lateral connections within individual layers in the brain that iteratively act upon the current state of the layer to update it. This mechanism allows the brain to make fine grained predictions on complex tasks. A characteristic attribute of this mechanism is the recursive application of the lateral connections which can be thought of as shared weights in a recurrent model. The above views suggest that it is desirable to have deep network models that perform parameter sharing in order to make the iterative inference view complete.

## 3 ITERATIVE INFERENCE IN RESNETS

Our goal in this section is to formalize the notion of iterative inference in Resnets. We study the properties of representations that residual blocks tend to learn, as a result of being additive in nature, in contrast to traditional compositional networks. Specifically, we consider Resnet architectures (see figure 1) where the first hidden layer is a convolution layer, which is followed by $L$ residual blocks which may or may not have *shortcut* connections in between residual blocks.

A residual block applied on a representation $\mathbf{h}_i$ transforms the representation as,

$$\mathbf{h}_{i+1} = \mathbf{h}_i + F_i(\mathbf{h}_i) \tag{1}$$

Consider $L$ such residual blocks stacked on top of each other followed by a loss function. Then, we can Taylor expand any given loss function $\mathcal{L}$ recursively as,

$$\mathcal{L}(\mathbf{h}_L) = \mathcal{L}(\mathbf{h}_{L-1} + F_{L-1}(\mathbf{h}_{L-1})) \tag{2}$$

$$= \mathcal{L}(\mathbf{h}_{L-1}) + F_{L-1}(\mathbf{h}_{L-1}).\frac{\partial \mathcal{L}(\mathbf{h}_{L-1})}{\partial \mathbf{h}_{L-1}} \tag{3}$$

$$+ \mathcal{O}(F_{L-1}^2(\mathbf{h}_{L-1}))$$

Here we have Taylor expanded the loss function around $\mathbf{h}_{L-1}$. We can similarly expand the loss function recursively around $\mathbf{h}_{L-2}$ and so on until $\mathbf{h}_i$ and get,

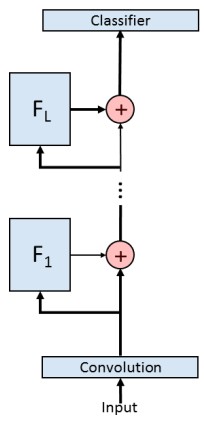

Figure 1: A typical residual network architecture.

$$\mathcal{L}(\mathbf{h}_L) = \mathcal{L}(\mathbf{h}_i) + \sum_{j=i}^{L-1} F_j(\mathbf{h}_j).\frac{\partial \mathcal{L}(\mathbf{h}_j)}{\partial \mathbf{h}_j} + \mathcal{O}(F_j^2(\mathbf{h}_j)) \tag{4}$$

Notice we have explicitly only written the first order terms of each expansion. The rest of the terms are absorbed in the higher order terms $\mathcal{O}(.)$. Further, the first order term is a good approximation when the magnitude of $F_j$ is small enough. In other cases, the higher order terms come into effect as well.

Thus in part, the loss equivalently minimizes the dot product between $F(\mathbf{h}_i)$ and $\frac{\partial \mathcal{L}(\mathbf{h}_i)}{\partial \mathbf{h}_i}$, which can be achieved by making $F(\mathbf{h}_i)$ point in the opposite half space to that of $\frac{\partial \mathcal{L}(\mathbf{h}_i)}{\partial \mathbf{h}_i}$. In other words, $\mathbf{h}_i + F(\mathbf{h}_i)$ approximately moves $\mathbf{h}_i$ in the same half space as that of $-\frac{\partial \mathcal{L}(\mathbf{h}_i)}{\partial \mathbf{h}_i}$. The overall training criteria can then be seen as approximately minimizing the dot product between these 2 terms along a path in the $\mathbf{h}$ space between $\mathbf{h}_i$ and $\mathbf{h}_L$ such that loss gradually reduces as we take steps from $\mathbf{h}_i$ to $\mathbf{h}_L$. The above analysis is justified in practice, as Resnets' top layers output $F_j$ has small magnitude (Greff et al., 2016), which we also report in Fig. 2.

Given our analysis we formalize iterative inference in Resnets as moving down the energy (loss) surface. It is also worth noting the resemblance of the function of a residual block to stochastic gradient descent. We make a more formal argument in the appendix.

## 4 EMPIRICAL ANALYSIS

Experiments are performed on CIFAR-10 (Krizhevsky & Hinton, 2009) and CIFAR-100 (see appendix) using the original Resnet architecture He et al. (2016b) and two other architectures that we

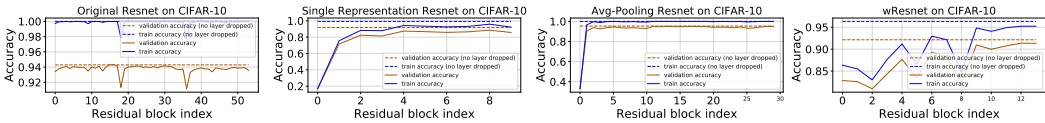

Figure 2: Average ratio of $\ell^2$ norm of output of residual block to the norm of the input of residual block for (left to right) original Resnet, single representation Resnet, avg-pooling Resnet, and wideResnet on CIFAR-10. (Train and validation curves are overlapping.)

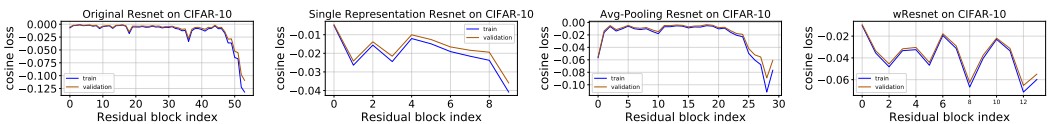

Figure 3: Final prediction accuracy when individual residual blocks are dropped for (left to right) original Resnet, single representation Resnet, avg-pooling Resnet, and wideResnet on CIFAR-10.

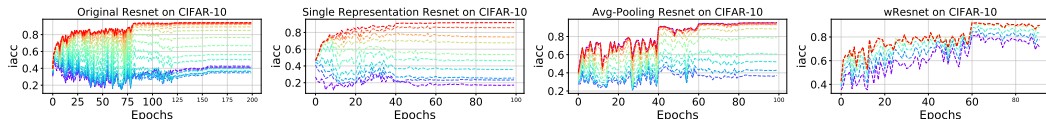

Figure 4: Average cos loss between residual block $F(\mathbf{h}_i)$ and $\frac{\partial \mathcal{L}(\mathbf{h}_i)}{\partial \mathbf{h}_i}$ for (left to right) original Resnet, single representation Resnet, avg-pooling Resnet, and wideResnet on CIFAR-10.

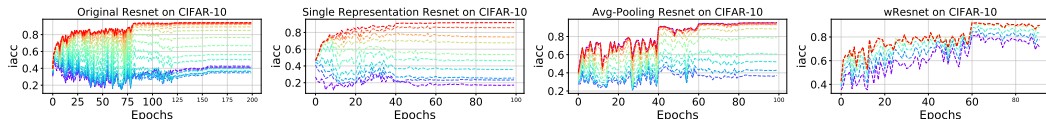

Figure 5: Prediction accuracy when plugging classifier after hidden states in the last stage of Resnets(if any) during training for (left to right) original Resnet, single representation Resnet, avg-pooling Resnet, and wideResnet on CIFAR-10. (Blue to red spectrum denotes lower to higher residual blocks)

introduce for the purpose of our analysis (described below). Our main goal is to validate that residual networks perform iterative refinement as discussed above, showing its various consequences. Specifically, we set out to empirically answer the following questions:

- Do residual blocks in Resnets behave similarly to each other or is there a distinction between blocks that perform iterative refinement vs. representation learning?
- Is the cosine between $\frac{\partial \mathcal{L}(\mathbf{h}_i)}{\partial \mathbf{h}_i}$ and $F_i(\mathbf{h}_i)$ negative in residual networks?
- What kind of samples do residual blocks target?
- What happens when layers are shared in Resnets?

**Resnet architectures**: We use the following four architectures for our analysis:

1. Original Resnet-110 architecture: This is the same architecture as used in He et al. (2016b) starting with a $3 \times 3$ convolution layer with 16 filters followed by 54 residual blocks in three different stages (of 18 blocks each with 16, 32 and 64 filters respectively) each separated by a shortcut connections ($1 \times 1$ convolution layers that allow change in the hidden space dimensionality) inserted after the $18^{th}$ and $36^{th}$ residual blocks such that the 3 stages have hidden space of height-width $32 \times 32$, $16 \times 16$ and $8 \times 8$. The model has a total of $1,742,762$ parameters.

2. Single representation Resnet: This architecture starts with a $3 \times 3$ convolution layer with 100 filters. This is followed by 10 residual blocks such that all hidden representations have the same height and width of $32 \times 32$ and 100 filters are used in all the convolution layers in residual blocks as well.

3. Avg-pooling Resnet: This architecture repeats the residual blocks of the single representation Resnet (described above) three times such that there is a $2 \times 2$ average pooling layer after each set of 10 residual blocks that reduces the height and width after each stage by half. Also, in contrast to single representation architecture, it uses 150 filters in all convolution layers. This is followed by the classification block as in the single representation Resnet. It has $12,201,310$ parameters. We call this architecture the avg-pooling architecture. We also ran experiments with max pooling instead of average pooling but do not report results because they were similar except that max pool acts more non-linearly compared with average pooling, and hence the metrics from max pooling are more similar to those from original Resnet.

4. Wide Resnet: This architecture starts with a $3 \times 3$ convolution layer followed by 3 stages of four residual blocks with 160, 320 and 640 number of filters respectively, and $3 \times 3$ kernel size in all convolution layers. This model has a total of 45,732,842 parameters.

**Experimental details:** For all architectures, we use *He-normal* weight initialization as suggested in He et al. (2015), and biases are initialized to 0. For residual blocks, we use BatchNorm→ReLU→Conv→BatchNorm→ReLU→Conv as suggested in He et al. (2016b). The classifier is composed of the following elements: BatchNorm→ReLU→AveragePool(8,8)→Flatten→Fully-Connected-Layer(#classes)→Softmax. This model has $1,829,210$ parameters. For all experiments for single representation and pooling Resnet architectures, we use SGD with momentum 0.9 and train for 200 epochs and 100 epochs (respectively) with learning rate 0.1 until epoch 40, 0.02 until 60, 0.004 until 80 and 0.0008 afterwards. For the original Resnet we use SGD with momentum 0.9 and train for 300 epochs with learning rate 0.1 until epoch 80, 0.01 until 120, 0.001 until 200, 0.00001 until 240 and 0.000011 afterwards. We use data augmentation (horizontal flipping and translation) during training of all architectures. For the wide Resnet architecture, we train the model with with learning rate 0.1 until epoch 60 and 0.02 until 100 epochs.

Note: All experiments on CIFAR-100 are reported in the appendix. In addition, we also record the metrics reported in sections 4.1 and 4.2 as a function of epochs (shown in the appendix due to space limitations). The conclusions are similar to what is reported below.

### 4.1 Cosine Loss of Residual Blocks

In this experiment we directly validate our theoretical prediction about Resnets minimizing the dot product between gradient of loss and block output. To this end compute the cosine loss $\frac{F_i(\mathbf{h}_i) \cdot \frac{\partial \mathcal{L}(\mathbf{h}_i)}{\partial \mathbf{h}_i}}{\|F_i(\mathbf{h}_i)\|_2 \|\frac{\partial \mathcal{L}(\mathbf{h}_i)}{\partial \mathbf{h}_i}\|_2}$. A negative cosine loss and small $F_i(.)$ together suggest that $F_i(.)$ is refining features by moving them in the half space of $-\frac{\partial \mathcal{L}(\mathbf{h}_i)}{\partial \mathbf{h}_i}$, thus reducing the loss value for the corresponding data samples. Figure 4 shows the cosine loss for CIFAR-10 on train and validation sets. These figures show that cosine loss is consistently negative for all residual blocks but especially for the higher residual blocks. Also, notice for deeper architectures (original Resnet and pooling Resnet), the higher blocks achieve more negative cosine loss and are thus more iterative in nature. Further, since the higher residual blocks make smaller changes to representation (figure 2), the first order Taylor's term becomes dominant and hence these blocks effectively move samples in the half space of the negative cosine loss thus reducing loss value of prediction. This result formalizes the sense in which residual blocks perform iterative refinement of features– move representations in the half space of $-\frac{\partial \mathcal{L}(\mathbf{h}_i)}{\partial \mathbf{h}_i}$.

### 4.2 Representation Learning vs. Feature Refinement

In this section, we are interested in investigating the behavior of residual layers in terms of representation learning vs. refinement of features. To this end, we perform the following experiments.

1. $\ell^2$ ratio $\|F_i(\mathbf{h}_i)\|_2/\|\mathbf{h}_i\|_2$: A residual block $F_i(.)$ transforms representation as $\mathbf{h}_{i+1} = \mathbf{h}_i + F_i(\mathbf{h}_i)$. For every such block in a Resnet, we measure the $\ell^2$ ratio of $\|F_i(\mathbf{h}_i)\|_2/\|\mathbf{h}_i\|_2$ averaged across samples. This ratio directly shows how significantly $F_i(.)$ changes the representation $\mathbf{h}_i$; a large change can be argued to be a necessary condition for layer to perform representation learning. Figure 2 shows the $\ell^2$ ratio for CIFAR-10 on train and validation sets. For single representation

Resnet and pooling Resnet, the first few residual blocks (especially the first residual block) changes representations significantly (up to twice the norm of the original representation), while the rest of the higher blocks are relatively much less significant and this effect is monotonic as we go to higher blocks. However this effect is not as drastic in the original Resnet and wide Resnet architectures which have two $1 \times 1$ (shortcut) convolution layers, thus adding up to a total of 3 convolution layers in the main path of the residual network (notice there exists only one convolution layer in the main path for the other two architectures). This suggests that residual blocks in general tend to learn to refine features but in the case when the network lacks enough compositional layers in the main path, lower residual blocks are forced to change representations significantly, as a proxy for the absence of compositional layers. Additionally, small $\ell^2$ ratio justifies first order approximation used to derive our main result in Sec. 3.

2. Effect of dropping residual layer on accuracy: We drop individual residual blocks from trained Resnets and make predictions using the rest of network on validation set. This analysis shows the significance of individual residual blocks towards the final accuracy that is achieved using all the residual blocks. Note, dropping individual residual blocks is possible because adjacent blocks operate in the same feature space. Figure 3 shows the result of dropping individual residual blocks. As one would expect given above analysis, dropping the first few residual layers (especially the first) for single representation Resnet and pooling Resnet leads to catastrophic performance drop while dropping most of the higher residual layers have minimal effect on performance. On the other hand, performance drops are not drastic for the original Resnet and wide Resnet architecture, which is in agreement with the observations in $\ell^2$ ratio experiments above.

In another set of experiments, we measure validation accuracy after individual residual block during the training process. This set of experiments is achieved by plugging the classifier right after each residual block in the last stage of hidden representation (i.e., after the last shortcut connection, if any). This is shown in figure 5. The figures show that accuracy increases very gradually when adding more residual blocks in the last stage of all architectures.

### 4.3    BORDERLINE EXAMPLES

In this section we investigate which samples get correctly classified after the application of a residual block. Individual residual blocks in general lead to small improvements in performance. Intuitively, since these layers move representations minimally (as shown by previous analysis), the samples that lead to these minor accuracy jump should be near the decision boundary but getting misclassified by a slight margin. To confirm this intuition, we focus on *borderline* examples, defined as examples that require less than $10\%$ probability change to flip prediction to, or from the correct class. We measure loss, accuracy and entropy over borderline examples over last $5$ blocks of the network using the network final classifier. Experiment is performed on CIFAR-10 using Resnet-110 architecture.

Fig 6 shows evolution of loss and accuracy on three groups of examples: *borderline* examples, *already correctly classified* and the whole dataset. While overall accuracy and loss remains similar across the top residual blocks, we observe that a significant chunk of *borderline* examples gets corrected by the immediate next residual block. This exposes the qualitative nature of examples that these feature refinement layers focus on, which is further reinforced by the fact that entropy decreases for all considered subsets. We also note that while train loss drops uniformly across layers, test sets loss *increases* after last block. Correcting this phenomenon could lead to improved generalization in Resnets, which we leave for future work.

### 4.4    UNROLLING RESIDUAL NETWORK

A fundamental requirement for a procedure to be truly iterative is to apply the same function. In this section we explore what happens when we unroll the last block of a trained residual network for more steps than it was trained for. Our main goal is to investigate if iterative inference generalizes to more steps than it was trained on. We focus on the same model as discussed in previous section, Resnet-110, and unroll the last residual block for 20 extra steps. Naively unrolling the network leads to activation explosion (we observe similar behavior in Sec. 4.5). To control for that effect, we added a scaling factor on the output of the last residual blocks. We hypothesize that controlling the scale limits the drift of the activation through the unrolled layer, i.e. they remains in a given neighbourhood on which the network is well behaved. Similarly to Sec. 4.3 we track evolution of

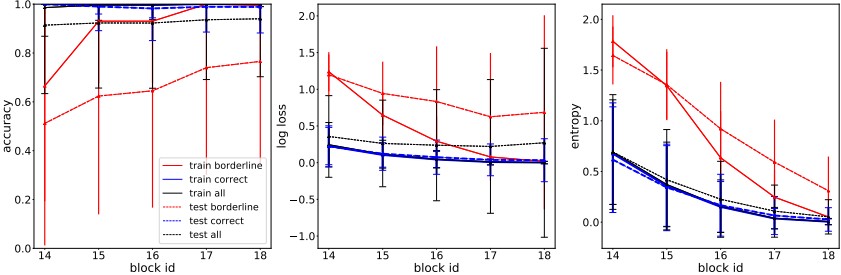

Figure 6: Accuracy, loss and entropy for last 5 blocks of Resnet-110. Performance on *bordeline* examples improves at the expense of performance (loss) of already correctly classified points (*correct*). This happens because last block output is encouraged by training to be negatively correlated (around $-0.1$ cosine) with gradient of the loss.

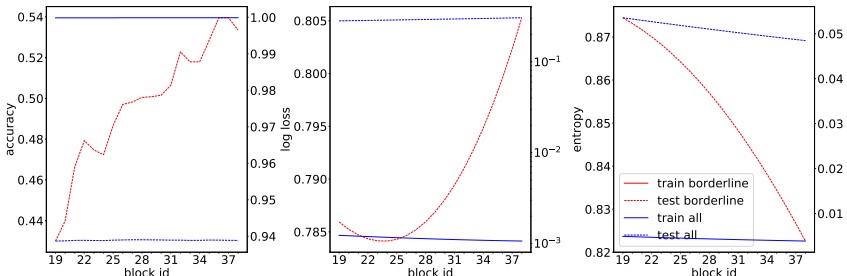

Figure 7: Accuracy, loss and entropy for Resnet-110 with last block unrolled for 20 *additional* steps (with appropriate scaling). *Borderline* examples are corrected and overall performance accuracy improves. Note different scales for train and test. Curves are averaged over 4 runs.

loss and accuracy on three groups of examples: *borderline* examples, *already correctly classified* and the whole dataset. Experiments are repeated 4 times, and results are averaged.

We first investigate how unrolling blocks impact loss and accuracy. Loss on train set improved uniformly from $0.0012$ to $0.001$, while it increased on test set. There are on average 51 borderline examples in test set[2], on which performance is improved from $43\%$ to $53\%$, which yields slight improvement in accuracy on test set. Next we shift our attention to cosine loss. We observe that cosine loss remains negative on the first two steps without rescaling, and all steps after scaling. Figure 7 shows evolution of loss and accuracy on the three groups of examples: *borderline* examples, *already correctly classified* and the whole dataset. Cosine loss and $\ell^2$ ratio for each block are reported in Appendix E.

To summarize, unrolling residual network to more steps than it was trained on improves both loss on train set, and maintains (in given neighbourhood) negative cosine loss on both train and test set.

## 4.5 SHARING RESIDUAL LAYERS

Our results suggest that top residual blocks should be shareable, because they perform similar iterative refinement. We consider a shared version of Resnet-110 model, where in each stage we share all the residual blocks from the $5^{th}$ block. All shared Resnets in this section have therefore a similar number of parameters as Resnet-38. Contrary to (Liao & Poggio, 2016) we observe that naively sharing the higher (iterative refinement) residual blocks of a Resnets in general leads to bad performance[3] (especially for deeper Resnets).

First, we compare the unshared and shared version of Resnet-110. The shared version uses approximately 3 times less parameters. In Fig. 8, we report the train and validation performances of the Resnet-110. We observe that naively sharing parameters of the top residual blocks leads both to

---

[2] All examples from train set have confident predictions by last block in the residual network.

[3] (Liao & Poggio, 2016) compared shallow Resnets with shared network having more residual blocks.

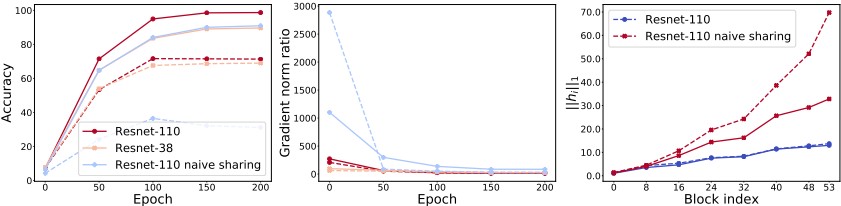

Figure 8: Resnet-110 with naively shared top 13 layers of each block compared with unshared Resnet-38. Left plot present training and validation curves, shared Resnet-110 heavily overfits. In the right plot we track gradient norm ratio between first block in first and last stage of resnet (i.e. $r = ||\frac{\partial L}{\partial h_1}||/\frac{\partial L}{\partial h_{1+2n}}||$). Significantly larger ratio in the naive sharing model suggests, that the overfitting is caused by early layers dominating learning. Metrics are tracked on train (solid line) and validation data (dashed line)
.

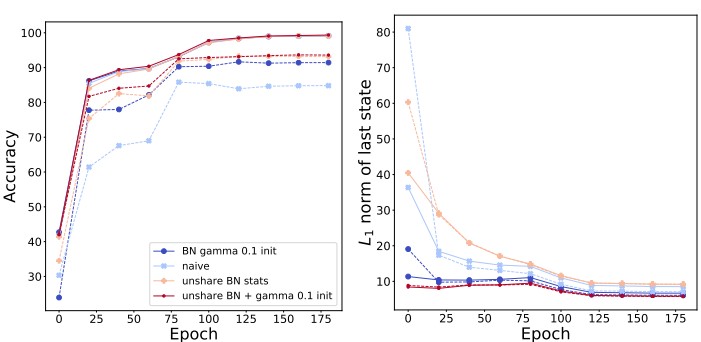

Figure 9: Ablation study of different strategies to remedy sharing leading to overfitting phenomenon in Residual Networks. Left figure shows effect on training and test accuracy. Right figure studies norm explosion. All components are important, but it is most crucial to unshare BN statistics.

overfitting (given similar training accuracy, the shared Resnet-110 has significantly lower validation performances) and underfitting (worse training accuracy than Resnet-110). We also compared our shared model with a Resnet-38 that has a similar number of parameters and observe worse validation performances, while achieving similar training accuracy.

We notice that sharing layers make the layer activations explode during the forward propagation at initialization due to the repeated application of the same operation (Fig 8, right). Consequently, the norm of the gradients also explodes at initialization (Fig. 8, center).

To address this issue we introduce a variant of recurrent batch normalization (Cooijmans et al., 2016), which proposes to initialize $\gamma$ to 0.1 and unshare statistics for every step. On top of this strategy, we also unshare $\gamma$ and $\beta$ parameters. Tab. 1 shows that using our strategy alleviates explosion problem and leads to small improvement over baseline with similar number of parameters. We also perform an ablation to study, see Figure. 9 (left), which show that all additions to naive strategy are necessary and drastically reduce the initial activation explosion. Finally, we observe a similar trend for cosine loss, intermediate accuracy, and $\ell^2$ ratio for the shared Resnet as for the unshared Resnet discussed in the previous Sections. Full results are reported in Appendix D.

Unshared Batch Normalization strategy therefore mitigates this exploding activation problem. This problem, leading to exploding gradient in our case, appears frequently in recurrent neural network. This suggests that future unrolled Resnets should use insights from research on recurrent networks optimization, including careful initialization (Henaff et al., 2016) and parametrization changes (Hochreiter & Schmidhuber, 1997).

| Model | CIFAR10 | CIFAR100 | Parameters |
|---|---|---|---|
| Resnet-32 | 1.53 / 7.14 | 12.62 / 30.08 | $467k$-$473k$ |
| Resnet-38 | 1.20 / 6.99 | 10.04 / 29.66 | $565k$-$571k$ |
| Resnet-110-UBN | 0.63 / **6.62** | 7.75 / 29.94 | $570k$-$576k$ |
| Resnet-146-UBN | 0.68 / 6.82 | 7.21 / 29.49 | $573k$-$579k$ |
| Resnet-182-UBN | 0.48 / 6.97 | 6.42 / **29.33** | $576k$-$581k$ |
| Resnet-56 | 0.58 / 6.53 | 5.19 / 28.99 | $857k$-$863k$ |
| Resnet-110 | 0.22 / 6.13 | 1.26 / 27.54 | $1734k$-$1740k$ |

Table 1: Train and test error of Resnet sharing top layers blocks (while using unshared both statistics and $\beta$, $\gamma$ in Batch Normalization) denoted as UBN (Unshared Batch Normalization) compared to baseline Resnet of varying depth. Training Resnet with unrolled layers can bring additional gain of 0.3%, while adding marginal amount of extra parameters. Runs are repeated 4 times.

## 5 CONCLUSION

Our main contribution is formalizing the view of iterative refinement in Resnets and showing analytically that residual blocks naturally encourage representations to move in the half space of negative loss gradient, thus implementing a gradient descent in the activation space (each block reduces loss and improves accuracy). We validate theory experimentally on a wide range of Resnet architectures.

We further explored two forms of sharing blocks in Resnet. We show that Resnet can be unrolled to more steps than it was trained on. Next, we found that counterintuitively training residual blocks with shared blocks leads to overfitting. While we propose a variant of batch normalization to mitigate it, we leave further investigation of this phenomena for future work. We hope that our developed formal view, and practical results, will aid analysis of other models employing iterative inference and residual connections.

## ACKNOWLEDGEMENTS

We acknowledge the computing resources provided by ComputeCanada and CalculQuebec. SJ was supported by Grant No. DI 2014/016644 from Ministry of Science and Higher Education, Poland. DA was supported by IVADO.

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

# Appendices

## A  FURTHER ANALYSIS

### A.1  A SIDE-EFFECT OF MOVING IN THE HALF SPACE OF $-\frac{\partial \mathcal{L}(\mathbf{h}_o)}{\partial \mathbf{h}_o}$

Let $\mathbf{h}_o = \mathbf{W}\mathbf{x} + \mathbf{b}$ be the output of the first layer (convolution) of a ResNet. In this analysis we show that if $\mathbf{h}_o$ moves in the half space of $-\frac{\partial \mathcal{L}(\mathbf{h}_o)}{\partial \mathbf{h}_o}$, then it is equivalent to updating the parameters of the convolution layer using a gradient update step. To see this, consider the change in $\mathbf{h}_o$ from updating parameters using gradient descent with step size $\eta$. This is given by,

$$\Delta \mathbf{h}_o = (\mathbf{W} - \eta \frac{\partial \mathcal{L}}{\partial \mathbf{W}})\mathbf{x} + (\mathbf{b} - \eta \frac{\partial \mathcal{L}}{\partial \mathbf{b}}) - (\mathbf{W}\mathbf{x} + \mathbf{b}) \tag{5}$$

$$= -\eta \frac{\partial \mathcal{L}}{\partial \mathbf{W}}\mathbf{x} - \eta \frac{\partial \mathcal{L}}{\partial \mathbf{b}} \tag{6}$$

$$= -\eta \frac{\partial \mathcal{L}}{\partial \mathbf{h}_o} \left( \frac{\partial \mathbf{h}_o}{\partial \mathbf{W}}\mathbf{x} + \frac{\partial \mathbf{h}_o}{\partial \mathbf{b}} \right) \tag{7}$$

$$= -\eta \frac{\partial \mathcal{L}}{\partial \mathbf{h}_o} \left( \|\mathbf{x}\|^2 + 1 \right) \tag{8}$$

$$\propto -\frac{\partial \mathcal{L}}{\partial \mathbf{h}_o} \tag{9}$$

Thus, moving $\mathbf{h}_o$ in the half space of $-\frac{\partial \mathcal{L}}{\partial \mathbf{h}_o}$ has the same effect as that achieved by updating the parameters $\mathbf{W}, \mathbf{b}$ using gradient descent. Although we found this insight interesting, we don't build upon it in this paper. We leave this as a future work.

## B  ANALYSIS ON CIFAR-100

Here we report the experiments as done in sections 4.2 and 4.1, for CIFAR-100 dataset. The plots are shown in figures 10, 11 and 12. The conclusions are same as reported in the main text for CIFAR-10.

## C  ANALYSIS OF INTERMEDIATE METRICS ON CIFAR-10 AND CIFAR-100

Here we plot the accuracy, cosine loss and $\ell^2$ ratio metrics corresponding to each individual residual block on validation during the training process for CIFAR-10 (figures 13, 14, 5) and CIFAR-100 (figures 15, 16, 17). These plots are recorded only for the residual blocks in the last space for each architecture (this is because otherwise the dimensions of the output of the residual block and the classifier will not match). In the case of cosine loss after individual residual block, this set of experiments is achieved by plugging the classifier right after each hidden representation and measuring the cosine between the gradient w.r.t. hidden representation and the corresponding residual block's output.

We find that the accuracy after individual residual blocks increases gradually as we move from from lower to higher residua blocks. Cosine loss on the other hand consistently remains negative for all architectures. Finally $\ell^2$ ratio tends to increase for residual blocks as training progresses.

## D  ITERATIVE INFERENCE IN SHARED RESNET

In this section we extend results from Sec. 4.5. We report cosine loss, intermediate accuracy, and $\ell^2$ ratio for naively shared Resnet in Fig. 19, and with unshared batch normalization in Fig. **??**.

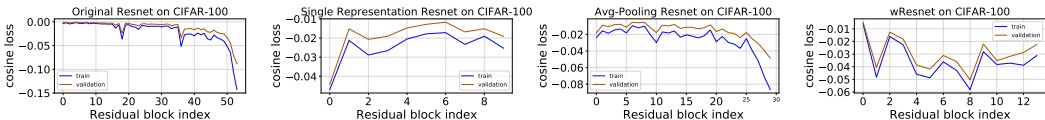

Figure 10: Average cos loss between residual block $F(\mathbf{h}_i)$ and $\frac{\partial \mathcal{L}(\mathbf{h}_i)}{\partial \mathbf{h}_i}$ for (left to right) original Resnet, single representation Resnet, avg-pooling Resnet, and wideResnet on CIFAR-100.

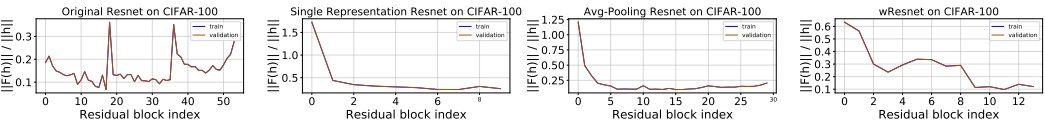

Figure 11: Average ratio of $\ell^2$ norm of output of residual block to the norm of the input of residual block for (left to right) original Resnet, single representation Resnet, avg-pooling Resnet, and wideResnet on CIFAR-100. (Train and validation curves are overlapping.)

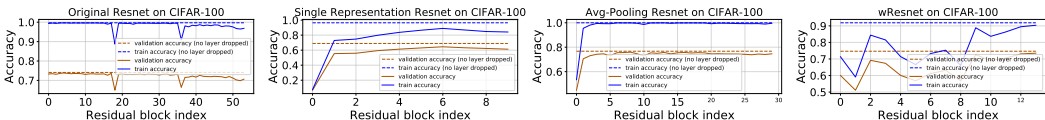

Figure 12: Final prediction accuracy when individual residual blocks are dropped for (left to right) original Resnet, single representation Resnet, avg-pooling Resnet, and wideResnet on CIFAR-100.

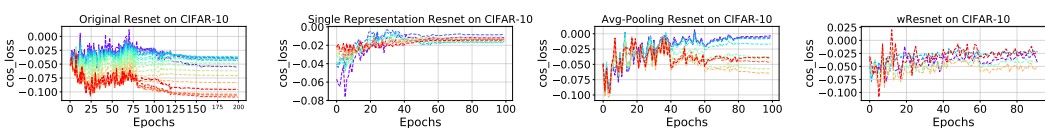

Figure 13: Average cos loss between residual block $F(\mathbf{h}_i)$ and $\frac{\partial \mathcal{L}(\mathbf{h}_i)}{\partial \mathbf{h}_i}$ during training for (left to right) original Resnet, single representation Resnet, avg-pooling Resnet, and wideResnet on CIFAR-10. (Blue to red spectrum denotes lower to higher residual blocks)

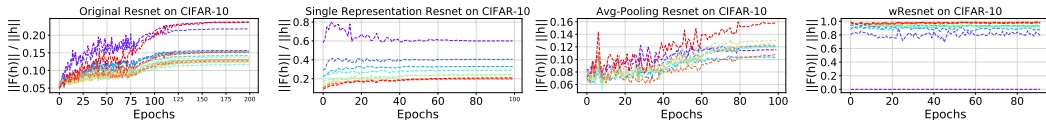

Figure 14: Average ratio of $\ell^2$ norm of output of residual block to the norm of the input of residual block during training for (left to right) original Resnet, single representation Resnet, avg-pooling Resnet, and wideResnet on CIFAR-10. (Blue to red spectrum denotes lower to higher residual blocks)

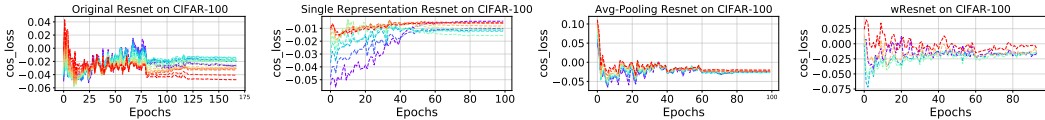

Figure 15: Average cos loss between residual block $F(\mathbf{h}_i)$ and $\frac{\partial \mathcal{L}(\mathbf{h}_i)}{\partial \mathbf{h}_i}$ during training for (left to right) original Resnet, single representation Resnet, avg-pooling Resnet, and wideResnet on CIFAR-100. (Blue to red spectrum denotes lower to higher residual blocks)

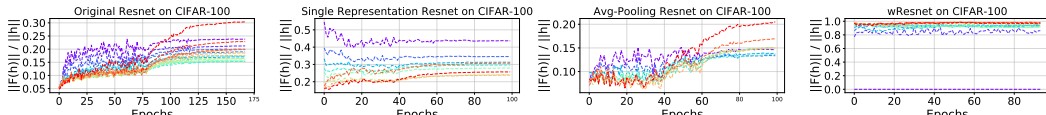

Figure 16: Average ratio of $\ell^2$ norm of output of residual block to the norm of the input of residual block during training for (left to right) original Resnet, single representation Resnet, avg-pooling Resnet, and wideResnet on CIFAR-100. (Blue to red spectrum denotes lower to higher residual blocks)

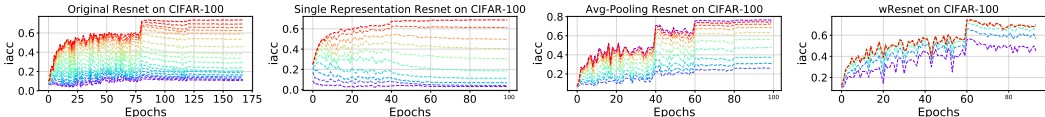

Figure 17: Prediction accuracy when plugging classifier after hidden states in the last stage of Resnets(if any) during training for (left to right) original Resnet, single representation Resnet, avg-pooling Resnet, and wideResnet on CIFAR-100. (Blue to red spectrum denotes lower to higher residual blocks)

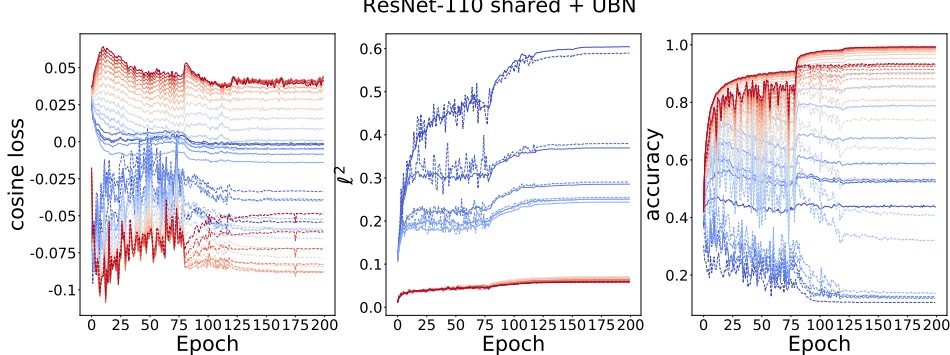

Figure 18: Cosine loss, $\ell^2$ ratio, and intermediate accuracy for shared Resnet-110 with unshared Batch Normalization (described in Sec. 4.5). Each curve represents different block in Resnet. Red is closest to output.

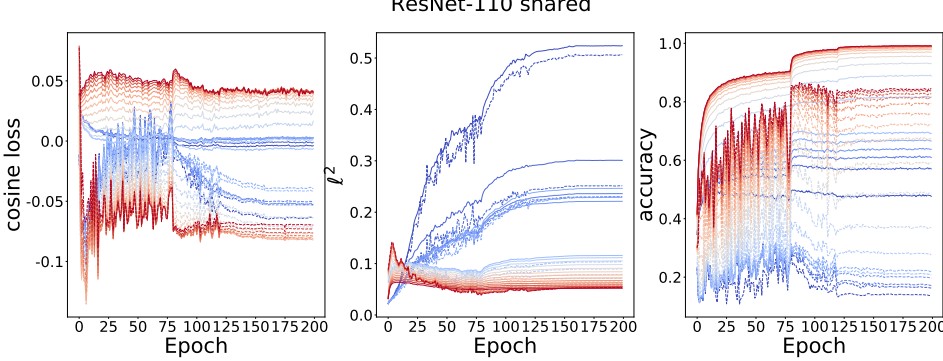

Figure 19: Cosine loss, $\ell^2$ ratio, and intermediate accuracy for naively shared Resnet-110. Each curve represents different block in Resnet. Red is closest to output.

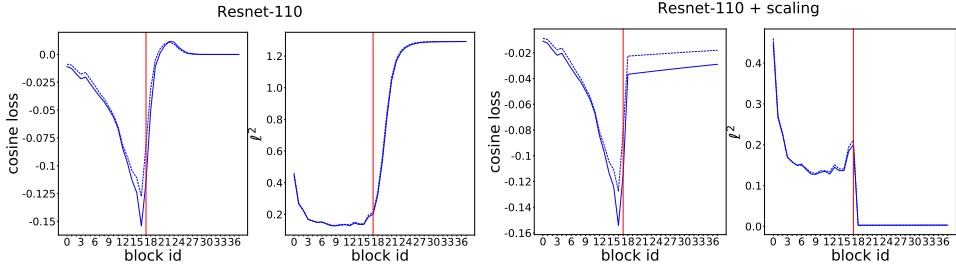

Figure 20: First figure show that cosine loss in Resnet-110 after unrolling generalizes to more steps than it was trained on. Second plot shows evolution of $\ell^2$ ratio for Resnet-110. Third plot reports cosine loss Resnet-110 with scaled version of final block, as considered in Sec. 4.4. Rightmost plots reports $\ell^2$ ratio for scaled Resnet-110. Vertical line in plots indicates number of steps network was trained on.

# E UNROLLING RESIDUAL NETWORKS

In this section we report additional results for unrolling residual network. Figure 20 shows evolution of cosine loss an $\ell^2$ ratio for Resnet-110 with unrolled last block for 20 additional steps.

