# OpenReview forum: "Residual Connections Encourage Iterative Inference"
_ICLR.cc/2018/Conference — Accept (Poster)_

### Official Review · AnonReviewer2 · 2017-11-26

**Rating:** 6
**Confidence:** 3

**Review:**

This paper investigates residual networks (ResNets) in an empirical way. The authors argue that shallow layers are responsible for learning important feature representations, while deeper layers focus on refining the features. They validate this point by performing a series of lesion study on ResNet.

Overall, the experiments and discussions in the first part of Section 4.2 and 4.3 appears to be interesting, while other observations are not quite surprising. I have two questions:
1)	What is the different between the layer-dropping experiment in sec 4.2 and that in [Veit, et al, Residual networks are exponential ensembles of relatively shallow networks] ? What is the main point here?
2)	I don't quite understand the first paragraph of sec 4.5. Could you elaborate more on this?

---

> ### Author Response · Authors · 2017-12-15
> **Response to AnonReviewer2**
>
> We thank reviewer for his remarks, and positive assessment.
>
> In his first point, reviewer asks what is difference between our 4.2 and Veit et al. We cite Veit et al, and extend his observations. Our novel observation is that blocks in residual network have different function, and only subset of blocks focus on iterative inference. More specifically, some blocks have large l2 ratio (ratio of output to input norms for given block), and cannot be dropped without drastic effect on performance. This allows us to specify concretely in what sense residual network performs iterative inference. We made edits in text to clarify this.
>
> In his second point reviewer requests clarification on first paragraph of 4.5. First paragraph of 4.5 reads: “Given the iterative inference view, we now study the effects of sharing residual blocks. Contrary to (Liao & Poggio, 2016) we observe that naively sharing the higher (iterative refinement) residual blocks of a Resnets in general leads to overfitting (especially for deeper Resnets).”.First, we say that our results suggest that residual network perform iterative inference, and that top blocks are performing similar function (feature refinement), there it is plausible that top blocks in residual network should be shareable. However, during this investigation, we report a surprising observation that has not been made before (Liao & Poggio tested relatively small ResNets) that when we share layers of residual network, it leads to drastic overfitting. In Fig.8 we compare Resnet-110-shared and Resnet-32, where Resnet-110-shared has same number of parameters as Resnet-32. We observe strong overfitting (train accuracy remains the same, while validation accuracy is much lower for Resnet110). We made edits in text to clarify this first paragraph.

---

### Official Review · AnonReviewer1 · 2017-11-28
**Evidences are not solid enough**

**Rating:** 5
**Confidence:** 5

**Review:**

The author unveils some properties of the resnets, for example, the cosine loss and l2 ratio of the layers.
I think the author should place more focus to study "real" iterative inference with shared parameters rather than analyzing original resnets.

In resnet without sharing parameters, it is quite ambiguous to say whether it is doing representation learning or iterative refinement.

1. The cosine loss is not meaningful in the sense that the classification layer is trained on the output of the last residual block and fixed. Moving the classification layer to early layers will definitely result in accuracy loss. Even in non-residual network, we can always say that the vector h_{i+1} - h_i is refining h_i towards the negative gradient direction. The motivation of iterative inference would be to generate a feature that is easier to classify rather than to match the current fixed classifier. Thus the final classification layer should be retrained for every addition or removal of residual blocks.

2. The l2 ratio. The l2 ratio is small for higher residual layers, I'm not sure how much this phenomenon can prove that resnet is actually doing iterative inference.

3. In section 4.4 it is shown that unrolling the layers can improve the performance of the network. However, the same can be achieved by adding more unshared layers. I think the study should focus more on whether shared or unshared is better.

4. Section 4.5 is a bit weak in experiments, my conclusion is that currently it is still limited by batch normalization and optimization, the evidence is still not strong enough to show that iterative inference is advantageous / disadvantageous.

The the above said, I think the more important thing is how we can benefit from iterative inference interpretation, which is relatively weak in this paper.

---

> ### Author Response · Authors · 2017-12-15
> **Response to AnonReviewer1 2/2**
>
> In his second point, reviewer asks why small l2 ratio in higher layers can be seen as a proof for iterative inference. Small l2 ratio suggests that 1st order taylor expansion is accurate. Decreasing l2 ratio from lower to higher layers further supports iterative inference, because model descending down the loss surface should converge towards the end. We made edits in revision to clarify these points.
>
> In his third point, reviewer mentions that it is not clear from 4.4 if adding shared or unshared layers is better. Section 4.5 is devoted to this question and concludes that shared layers do not lead to same gains as unshared. In particular, we observe that naively sharing parameters of the top residual blocks  leads both to underfitting (worse training accuracy than its unshared counterpart) and overfitting (given similar training accuracy, the model with shared parameters model has significantly lower validation performances).
> This issue appears to be related to the activation explosion we observe in the shared Resnet model. We believe that developing better strategy to control the activation norm increase through the layer could help addressing this issue.
>
> In his fourth point, reviewer remarks results of 4.5 are limited by optimization and using batch normalization. We agree in this sense, that optimization of shared residual network seems intrinsically difficult. Given our results from previous sections, we believe that bridging these difficulties in training, will result in a strong shared residual network.
>
> Overall, our novelty is in showing one specific, formal way in which residual networks perform iterative inference, and then exploring it in both unshared, and shared case (both by training shared network, as well as unrolling unshared network to more steps than it was trained on).

---

> ### Author Response · Authors · 2017-12-15
> **Response to AnonReviewer1 1/2**
>
> We thank reviewer 2 for his thoughtful review. To address the comments we added new plots to paper (2 sections in Appendix with references in main text), and revised text to clarify mentioned points.
>
> We would like to open by clarifying our definition of iterative inference. Iterative inference (as specified in “Formalizing iterative inference section”) is defined in our paper as descending the loss surface of the network that is specified by current set of parameters, rather than finding generally better features. Our contribution is showing that residual network maximizes the alignment of block output with steepest direction in the aforementioned loss surface, specified by the network current parameters. The usage of fixed classifier is therefore justifiedby our Taylor expansion. We made these points clearer in revised version, and also added more detailed math derivation.
>
> The central objection of reviewer is our focus on iterative inference without shared weights. Our objective is to study a form of iterative inference (as specified above) implemented by regular residual networks. However, we devoted large section of paper to shared residual network. We fully agree there is room for doubt if some of the results transfer to true iterative (shared) residual network, and we thank reviewer for this remark. To address this, we plot cosine loss, l2 ratio, and intermediate accuracy on shared residual networks, on which we observe very similar trends to unshared residual network. We posted them anonymously here: https://ibb.co/hd1F9m (red is close to output, dotted is on validation set), and also included them in the revision in appendix.
>
> Now, we respond in detail to each point in turn.
>
> In his first point, reviewer raises objection to using fixed classifier. We would like to stress that our definition of iterative inference is descending down the network loss surface here defined by fixed classifier. We made this point clearer in the revision. We also include plot of all the metrics for “true” iterative inference in shared residual network, https://ibb.co/hd1F9m (red is close to output, dotted is on validation set), also added to Appendix.
>
> Next objection is that trivially non-residual network could be seen as revising h_{t+1} towards good accuracy, and therefore cosine loss is not a meaningful metric. We are in agreement that in a non-residual network we do expect increase in accuracy. However, we do not expect from non-residual network:
> Layer output to be aligned with gradient of loss with respect of hidden state, and perform small iterative refinement. Layer output always should increase accuracy as stated by reviewer, but this is very different from small iterative steps that are aligned with dL/dh.
> Layer to  generalize when applied to more steps than it was trained (Sec 4.4), both on train and test distribution
> Final layers to focus only on borderline examples (as specified in text, examples that are close to being either correctly classified, or misclassified)
> , which are non-trivial things we report. To further support generalization to more steps we revised text to highlight maintained negative cos loss and reduction of loss on training set. We plot evolution of cosine loss for unrolled steps in appendix: https://ibb.co/f64YC6. To further support claim about iterative refinement, we plot l2 ratio for experiments in 4.3, 4.4 (also included in https://ibb.co/f64YC6).

---

### Official Review · AnonReviewer3 · 2017-11-29
**Review of "Residual Connections Encourage Iterative Inference"**

**Rating:** 7
**Confidence:** 4

**Review:**


This paper shows that residual networks can be viewed as doing a sort of iterative inference, where each layer is trained to use its “nonlinear part” to push its values in the negative direction of the loss gradient.  The authors demonstrate this using a Taylor expansion of a standard residual block first, then follow up with several experiments that corroborate this interpretation of iterative inference.  Overall the strength of this paper is that the main insight is quite interesting — though many people have informally thought of residual networks as having this interpretation — this paper is the first one to my knowledge to explain the intuition in a more precise way.

Some weaknesses of the paper on the other hand — some of the parts of the paper (e.g. on weight sharing) are only somewhat related to the main topic of the paper. In fact, the authors moved the connection to SGD to the appendix, which I thought would be *more* related.   Additionally, parts of the paper are not as clearly written as they could be and lack rigor.  This includes the mathematical derivation of the main insight — some of the steps should be spelled out more explicitly.  The explanation following is also handwavey despite claims to being formal.

Some other lower level thoughts:
* Regarding weight sharing for residual layers, I don’t understand why we can draw the conclusion that the initial gradient explosion is responsible for the lower generalization capability of the model with shared weights.  Are there other papers in literature that have shown this connection?
* The name “cosine loss” suggests that this function is actually being minimized by a training procedure, but it is just a value that is being plotted… perhaps just call it the cosine?
* I recommend that the authors also check out Figurnov et al CVPR 2017 ("Spatially Adaptive Computation Time for Residual Networks") which proposes an “adaptive” version of ResNet based on the intuition of adaptive inference.
* The plots in the later parts of the paper are quite small and hard to read.  They are also spaced together too tightly (horizontally), making it difficult to immediately see what each plot is supposed to represent via the y-axis label.
* Finally, the citations need to be fixed (use \citep{} instead of \cite{})

---

> ### Author Response · Authors · 2017-12-15
> **Response to AnonReviewer3**
>
> We thank reviewer for his positive assessment and useful feedbacks.
>
> In his first point reviewer remarks that sharing residual network is less interesting than studying SGD connection. We agree with reviewer 1 and will move back the connection to SGD in the main text in camera ready version. We do believe that studying weight sharing in residual network is important as well, because it implements ‘true’ iterative inference (i.e. where same function is applied).
>
> Next, reviewer suggests we should improve writing of some parts of paper including mathematical derivation. We address this remark in revision of the paper, by making the derivation more explicit (step from (3) to (4)).
>
> In the next point reviewer asks if interpretation in 4.5 that gradient explosion leads to overfitting is justified. We would like to clarify we observe both underfitting  (worse training accuracy of the shared model compared to the unshared model) and overfitting (given similar training accuracy, the Resnet with shared parameter has significantly lower validation performances). We also observe that the activations explode during the forward propagation of the Resnet with shared parameters due to the the repeated application of the same layer. By better controlling the activation norm increase using unshared batch normalization, we show that we can reduce the performance gap  between the Resnet with shared parameters and the Resnet with unshared parameters, for both training  accuracy and validation accuracy. We have updated the text in section 4.5 to clarify this point.
>
> We thank for reference.  We introduced suggested edits in revision. We agree that “cosine loss” name can be misleading. For camera ready we will change it.

---

### Author Response · Authors · 2018-01-05
**Explanation of changes from original submission**

To address reviewers comments we introduced following changes to the submission:
* We clarified derivation of main result, as suggested by Reviewer 1
* Clarified 4.3 and 4.5 to address reviewers various comments, including clearer description of overfitting when sharing Resnet blocks, rewording of first paragraph of 4.5, clarifying definition of borderline examples, and mentioned additional results in Appendix
* Clarified 4.4, underlying the goal was to show Resnet can generalize to more steps (on train), mentioned additional results in Appendix
* Clarified conclusions
* To address Reviewer 2 comment on iterative inference in shared Resnet, we added two sections in Appendix reporting metrics (cosine loss, accuracy, l1 ratio) on shared Resnet, and on the unrolled to more steps Resnet.
* Fixed some typos, and added some minor clarifications of other experiments

---

### Decision · Program_Chairs · 2018-01-29
**ICLR 2018 Conference Acceptance Decision**

**Decision:**

Accept (Poster)

**Comment:**

The paper presents an interesting view of ResNets and the findings should be of broad interest. R1 did not update their score/review, but I am satisfied with the author response, and recommend this paper for acceptance.